# Dynamic Spectrum Allocation Using Multi-Source Context Information in OpenRAN Networks

**DOI:** 10.3390/s22093515

**Published:** 2022-05-05

**Authors:** Łukasz Kułacz, Adrian Kliks

**Affiliations:** Institute of Radiocommunications, Poznan University of Technology, 60-965 Poznan, Poland; lukasz.kulacz@put.poznan.pl

**Keywords:** context information, dynamic spectrum sharing, open radio access network, radio service maps

## Abstract

Bearing in mind the stringent problem of limited and inefficiently used radio resources, a multi-source mechanism for the dynamic adjustment of occupied frequency bands is proposed. Instead of relying only on radio-related information, the system that collects data from various sources is discussed. Mainly, using the ubiquitous sources of information about the presence of users (such as city monitoring), it is possible to identify areas that have high or low expected traffic with high probabilities. Consequently, in low-traffic areas, it is not necessary to allocate all available spectrum resources while maintaining the quality of service. This leads to the improved spectral efficiency of the network. As the level of trust in certain information sources may differ among various operators, we propose to implement such functionality in the form of an application. Our contribution is a proposal for an algorithm that limits the use of radio resources through fuzzy and soft connections of multiple sources of contextual information. The simulation results presented in this paper show that it is possible to reduce the spectrum used with a slight and simultaneous reduction in user bitrate, which increases the spectral efficiency of the entire system. Hence, following the concept of an open radio access network, various policies for information merging may be specified.

## 1. Introduction

In view of the ever-growing demand for wireless services and of the increasing number of connected devices [1,2], the effective spectrum access mechanisms, and in a broader sense—the overall spectrum-usage efficiency, are very important aspects of current and future wireless systems. Taking into account the limited spectral resources [3,4] as well as the costs related to its utilization, great attention should be paid to the effective usage of all frequency resources that are already in use. However, in static spectrum allocation, the whole available bandwidth is typically used, which leads to the already mentioned inefficient spectrum utilization. Following the well-known concept of cognitive radio [5,6], it will be much more effective to allocate the band dynamically and release the unneeded spectrum resources to the publicly available pool. Such an approach offers the possibilities for flexible pricing and licensing [7] and for the allocation of the unused band to other services. Such a dynamic approach allows for an adjustment of the usage of resources to the changing environmental situation. A classic example is a network operating in a specific area (e.g., in a city), where during the night hours or during low-traffic hours in general, it would be possible to limit the bandwidth occupied by the network without degrading the performance metrics (e.g., the quality of services observed by users) or to offload some traffic to other bands [8,9,10]. However, in such an approach, the critical point is being able to reliably identify the areas which are less populated by active users. The typical and straightforward approach is to process all radio data available to the network operators in order to decide on the prospective wireless traffic generated in that region. The mobile network operator knows the approximate location of the users, and it is also aware of the user traffic profile, i.e., it knows the kind of traffic generated by a certain user, the traffic statistics as well as end-user habits (at what time the user generates the traffic the most often). Such knowledge allows for the prediction of the traffic that will be generated in a given area based on some historical reasoning and current observations. In consequence, it will be possible to allocate an amount of frequency resources that could guarantee smooth service delivery to the end-users (UE). However, the instantaneous situation in the network may be significantly different from the statistical models available to the operators. Thus, we affirm that the performance of the traffic-prediction algorithm (and therefore the frequency allocation algorithm) shall be supported by the access to various non-radio information sources in order to reflect the temporary changes in the network. For example, having in mind the high efficiency and high accuracy of currently available human detection algorithms, the access to public cameras (i.e., from city monitoring systems) could provide additional knowledge on a given number of end-users in a certain area at a certain time. In this paper, we evaluate the initial performance of such an approach, where non-radio and radio information are processed jointly for dynamic resource allocation. As such multi-source optimization requires advanced reasoning and decision making [11], which will be a subject for further research, in this work, we concentrate on the software-related aspect of this approach. Mainly following the recent and vivid concept of open radio access networks (ORAN) [12,13,14,15], where various RAN functionalities may be defined in terms of separate routines (called xAps or rAps) [16], we propose to implement the dynamic spectrum access function (DSA) as the xApp. In this approach, the policies on how various information sources shall be processed are defined freely in the form of a file. This is essential as various operators may have a different level of trust in various information sources and, in consequence, would like to give it more or less significance.

## 2. Merging Various Information Sources

### 2.1. Source I: Access Point as a Source of Information about Users

Let us consider a radio network with access points arranged on a regular grid and a set of end-users deployed with a uniform distribution over some area. Intentionally, in some places, the so-called hotspots are created, where many users generating potentially high traffic are located in a small area. Moreover, it is assumed that during the day, the overall number of users in that area changes. In consequence, the traffic demands (and hence the demands on the radio resources) will also change in time. As we demonstrate in this paper, by observing the true presence of users in the network at a certain time and in a given location, it will be possible to make a decision flexibly in order to limit the occupied band at some access points. We assume that such a cognitive radio functionality will be available to all these nodes, which follow the concept of ORAN; this is in contrast to traditional base stations, which always utilize the whole licensed frequency band. However, the ORAN-compliant base stations will be equipped with a decision-making mechanism, which will be available to the mobile network operator in the form of a downloadable and configurable application (xApp). The idea behind this application is to allow the mobile network operators to flexibly adjust the policies concerning the use of information from multiple sources about the presence of end-users in a certain location. Through the simple modification of file entries and the setup of the application, the importance and level of trust may be dynamically assigned to each information source, e.g., by setting various weighting coefficients associated with them. Moreover, various reasoning techniques may be specified in such a configuration file. However, before we go deeper into the multi-source decision-making process, let us discuss the prospective sources of contextual information that can be jointly used for better spectrum usage. In our case, we want to predict the coarse amount of traffic that will be generated in a given area—if the expected traffic is low, the rApp may deactivate unnecessary frequency resources and release them to the common pool. Thus, the focus is on collecting information about the density of users in a given location at a specific time. For simplicity, we will use a typical approach in radio service maps, where each piece of information is represented by a map with a regular grid.

### 2.2. Source II: Monitoring as a Source of Information about Users

The second type of information source that supports us in predicting the end-user locations and densities in the considered area can be all kinds of cameras widely deployed in cities nowadays. They can be found “at every step” at road intersections, parks, banks, etc., and their presence is primarily meant to improve people’s safety and enable the security services to react faster. By utilizing the people recognition software present in such surveillance systems, a vast amount of information about the density of people could be obtained. From the perspective of this paper, this is very important as the information from such a source will provide the density of both people using radio access to the network and those who do not currently use it. Thus, it can be effectively used to predict the potential new users who may be connected to the network and are starting transmissions. Unfortunately, the cameras are placed in different places, and their quality of observation depends on many factors. Nevertheless, assuming that there is an appropriate modeling of such data, it will certainly be possible to deduce from them where the areas of higher or lower user density are located. A map utilizing this type of data is created in a similar way to a map made of data from access points. The approximate number of users visible to the camera is placed at points on the map that indicate where the cameras are located. The remaining map points must be completed, e.g., by interpolation. Obviously, this type of information does not give a complete picture of the user density across the entire network. The coverage of cameras does not cover all areas; in addition, the system of recognizing the presence of people is not flawless.

### 2.3. Source III: Power Sensor as a Source of Information about Users

The third type of information on user density may come from the ubiquitous access points of other networks—most of all, the wireless local area network (WLAN) access points (such as IEEE 802.11 access points). In many places (especially in cities), there are publicly available access points to the radio network. Since this is a different network from the one discussed in this paper, we assume that the information about the users is made available only in the form of aggregated power received by these access points. By collecting such data and normalizing them, we can approximate the presence of users in the area. Again, such information is stored on the map at the exact location of the access point, and the remaining map points are supplemented.

### 2.4. Combined Data as a Source of Information about Users

Besides these three information sources, one may identify others and specify the ways they are incorporated into the whole system. However, let us treat these three information sources as a good example of combining multi-source information for dynamic spectrum access in ORAN-based networks. Thus, as discussed above, we assume that there are three kinds of information sources—the regular cellular network base stations, other access points (such as WLAN access points), and city cameras. As the situation in the network changes in time, we assume that for each information source, the corresponding map of user density is created for the specific time period. Moreover, as these information sources are of completely different kinds, the maps should be normalized before further processing. As there are numerous multi-source decision-making solutions, we suggest combining all the information into one final map by appropriately weighting the particular information sources. Of course, the method of combining maps, and ultimately, the selection of weighting factors for individual maps should be the subject of other research and optimization. Additionally, it is worth noting that the weights determined once should be constantly updated and adapted to changing conditions. Thus, let us assume a square area of size *D*, represented digitally in the form of a regular-grid map of equally sized small squares of size *d*, i.e., D=dN, where *N* is the number of tiles on the map in one dimension. Each tile will be indexed as (x,y), where x,y=1,…,N. For each information source $S_{i}$ (in our case i−1,2,3), one can create a map that stores all normalized values of the term associated with this information source. Let us denote the measured and normalized value from source $S_{i}$ at location (x,y) as $v_{\left(x,y\right)}^{S_{i}}$. Thus, the final combined map can be created by merging contributions from all information sources in the following way:(1)$$C_{\left(x,y\right)}=\sum_{i}\xi_{i}v_{\left(x,y\right)}^{S_{i}}\;,$$
where $C_{\left(x,y\right)}$ is the combined value at point (x,y), and $\xi_{i}$ is the weighting coefficient associated with the *i*-th information source.

### 2.5. Policy Defined Dynamic Spectrum Access

As discussed previously, how the information sources could be merged is subject to detailed analysis and reasoning. However, in our approach, the ultimate goal is to show the possibility of using the dynamic spectrum access algorithm implemented as a dedicated xApp in the ORAN environment. Thus, our ultimate goal is to provide the functionality to the mobile network operator to flexibly specify the ways in which they are using the spectrum in their network; in particular, they may relax unused spectral resources to the common pool. However, this decision will be made after some detailed reasoning based on information gathered from various sources (such as those described above). Various operators may have a different level of trust in various information sources; thus, they can modify the weighting coefficients used in (Equation 1). Moreover, the operators may specify the policies and how the spectrum is relaxed based on the current environmental conditions. To visualize the opportunities created by the ORAN concept, dedicated files (e.g., JSON, YAML) can be provided as input to the ORAN xApp, where all weighing coefficients can be specified and relaxing policies can be described as well [12]. In order to show the benefits of the softwarized spectrum sharing approach, in the following section, we evaluate the whole concept in detail.

## 3. Evaluation Process—Combined Map Creation

### 3.1. Considered Scenario

In order to test and confirm the ideas described earlier, the authors conducted several computer simulations. The list of parameters used during the simulation is presented in Table 1. The simulation considered a network fragment consisting of evenly spaced access points (placed on a regular mesh). As discussed, there are three sources of information deployed here. First, there are the base stations forming the regular grid, which track the number of connected users. Second, there are the city surveillance cameras, for which we assume that accuracy in human recognition reduces exponentially with distance. The cameras are placed in the analyzed area, in random positions with uniform distribution. The third source are the WLAN access points of other networks, which are also deployed randomly. An exemplary topology of the analyzed network is shown in Figure 1.

Most users (and people in general) are placed inside two rectangular areas: 40% (x∈<1;20>, y∈<2;4>) and 40% (x∈<13;14>, y∈<1;17>). The remaining 20% of users are located anywhere in the analyzed area. All items (within a single rectangle) are selected according to the uniform distribution. The chosen location of users reflects the situation where the majority of people are located on the two streets or sidewalks in the analyzed area (main streets).

The scenario of a single simulation consists of a certain number of days, where each day is divided into a certain number of time periods. In order to mimic the typical behavior of humans (walking, sitting in restaurants for some time, shopping, etc.), the number of users changes with the time of day (moment of the day).

The number of days represents simulation repetitions, that is, sets of random user locations. For the first $N_{days}$ days, all base stations use the entire available bandwidth. We treat it as the training phase when the data necessary to create user density maps (data of all three types) are collected. Furthermore, according to the proposed algorithm, the occupied bandwidth is limited in some base stations for the next $N_{days}$ days. A single iteration of the simulation consists of the following steps. First, the Euclidean distance between each user and each access point is computed. The power received by each user is then calculated based on the distance to the access points and the free-space loss model. Next, the users are assigned to the access point that received the highest power value. In the next step, the radio resources are distributed equally to each user within one access point. Finally, the signal-to-noise ratio and interference value are calculated, based on which the user bitrate is calculated (using Shannon’s formula).

### 3.2. Single Source and Combined Maps

Each type of user density information is used to create a final, combined user density map (at a given point in time). As mentioned earlier, the missing map points are approximated using interpolation. Moreover, all information sources are normalized. The example of achieved normalized maps that originated from the three above-mentioned sources at specific times are shown in Figure 2, Figure 3 and Figure 4, respectively. One may observe various accuracy levels achieved in these maps, which refer to, e.g., the density of information sources—there are fewer base stations present in the area compared to the number of cameras or WLAN access points. To combine all maps into one output, the normalized maps are added to each other using appropriate weights (see (Equation 1)); at this stage of the simulation, the weights of all maps are equal. An exemplary merged map is shown in Figure 5.

### 3.3. Algorithm for Limiting the Available Bandwidth in Access Points

When a final, combined map of user density is available, it is possible to decide to limit the used bandwidth in places with a low probability of user occurrence. Following the policies specified by the mobile network operators, each access point individually reads the point closest to its location from the user density map and adjusts the operating frequency range. To visualize the behavior of the proposed xApp, the authors selected the following illustrative policy with two fixed threshold values; in this example, these thresholds have been fixed, but in practice, these are subject to optimization. Two decision thresholds, $T_{1}$ and $T_{2}$, are applied: When the combined value is below $T_{1}$, the occupied frequency band is limited to 50% of the maximum bandwidth; when the combined value is greater than $T_{1}$ and lower than $T_{2}$, the band is limited to 75% of the maximum bandwidth. Otherwise, the access point will use 100% of the maximum bandwidth.

## 4. Simulation Results

To numerically evaluate the behavior of the system, we compare the selected performance metrics achieved for the traditional scenario (with no ORAN functionality), referred to as *reference*, with the situation when the dynamic spectrum xApp was applied to optimize the occupied band—denoted hereafter as *with DSA*. In this case, the reference scenario concerns the inability to control the spectrum that is dynamically used by the access points. The single repetition of the simulation consist of $N_{days}$ days of the *reference* scenario (learning time) and $N_{days}$ days of the *with DSA* scenario.

### 4.1. Single Simulation Repetition

An example of a single repetition of the simulation for a specific time is presented in Figure 6. The figure shows the cumulative distribution of user throughput in both scenarios. Additionally, changes in the respective throughput percentiles have been added to the figure. It can be noticed that in the *with DSA* scenario, we experience decreases in the bitrate of at least a few percent. However, as a result of the application of the DSA algorithm, we only use a total bandwidth of 520 MHz, whereas in the reference scenario, this value is 600 MHz (30 access points, 20 MHz each). To better assess the change above, it is also worth looking at the change in the value of spectral efficiency, which is expressed by the sum of all observed bitrates for $N_{days}$ days divided by the sum of the bandwidth used at that time. As a result of the application of the DSA algorithm, the spectral efficiency is 1.577 bit/Hz, whereas in the reference scenario, this value is 1.499 bit/Hz. Therefore, we observe an improvement in spectral efficiency at the level of 5.203%.

### 4.2. Multiple Simulation Repetitions

A single repetition of the simulation in this section was performed ten times in order to average the effect of the random position of the cameras and power sensors. Each repetition was performed for each value of the analyzed parameter. The charts show the minimum, average, and maximum values. These are functions that were used at different times of the day.

The first parameter to be simulated was the maximum number of users. The values of this parameter changed from 50 to 500 at step 50. This simulation was aimed at checking the validity of using the DSA algorithm in different user density conditions. The tenth percentile of user throughput is shown in Figure 7, and spectral efficiency is shown in Figure 8.

Again, one can draw the same conclusions as for a single simulation repetition; that is, by using the DSA algorithm, we reduce the user bitrate by a few percent, improving spectral efficiency and freeing up some spectral resources that other services can use.

Let us now analyze the impact of the thresholds $T_{1}$ and $T_{2}$. We change the values of $T_{1}$ within the range of 0–0.4, divide it into 30 equidistant values, and then set $T_{2}$ as 0.5 $T_{1}$. The obtained results are presented in Figure 9 for the 10th percentile of the user bitrate, and in Figure 10 for spectral efficiency. In the reference scenario, this value is more or less the same, while increasing the value of $T_{1}$ causes a decrease in throughput—especially the maximum values observed during the day. This is in line with expectations as the system capacity diminishes, and the capacity should not be reduced at peak times of the day. The values of $T_{1}=0$ for the DSA scenario mean there is de facto no reduction in the radio resource utilization as the detected user density would have to be less than 0 in order to reduce the bandwidth. Next, increasing the $T_{1}$ parameter improves spectral efficiency because the bandwidth used at the access points is reduced; however, the list of compromised users is the same, and their total bitrate is very similar. Nevertheless, it is worth paying attention to the spectrum efficiency values within the lowest user density case (line entitled “min—with DSA”). A local maximum of spectrum efficiency can be noticed there, where a further increase of parameter $T_{1}$ causes the deterioration of spectrum efficiency. However, in most cases (max and mean), there is a trend where the spectral efficiency increases with the increase of parameter $T_{1}$. This means that for most random user positions, the spectrum reduction algorithm has less impact on bitrate than on the total amount of spectrum used by the system. In a smaller number of cases, the spectral efficiency stops increasing with the increase of the $T_{1}$ parameter. This can be observed in a situation where there are many access points close to each other, where few users are present, and where each of these access points is very spectrum limiting. Moreover, a worse condition is where the bandwidth is limited by a higher user density, and a mistake is made due to the imprecise information gathered by the sensors. It is worth noting that the analyzed throughput charts show the 10th percentile of throughput, while the average value, median, and higher percentiles remain practically unchanged.

Finally, in Figure 11, the total bandwidth used by the network is shown. It can be noticed that increasing the parameter $T_{1}$ reduces the total band used in the system. All released resources may be used by other systems at the same time. By increasing the value of the T1 parameter, we modify the algorithm in such a way that it limits the use of the spectrum. Radio resource reduction is used for a higher density of detected users.

## 5. Discussion

In this work, the authors presented the possibility of using contextual data on user density from various sources in the ORAN-based networks. Depending on the expectations with regard to the effect of the algorithm, it is possible to improve the entire network’s spectral efficiency, thus keeping the users’ performance metrics at acceptable levels. The adjusted combination of various information sources (radio-based and non-radio-based) improves spectral efficiency as compared to the traditional scenario. Implementing the spectrum access method in the form of an xApp allows the mobile network operators to dynamically select the appropriate weighting coefficients, multi-source combining methods, and spectrum adjustment policy. All of these aspects are subject to further analysis; however, in this paper, we have shown the validity of this software-based spectrum allocation concept.

## Figures and Tables

**Figure 1 sensors-22-03515-f001:**
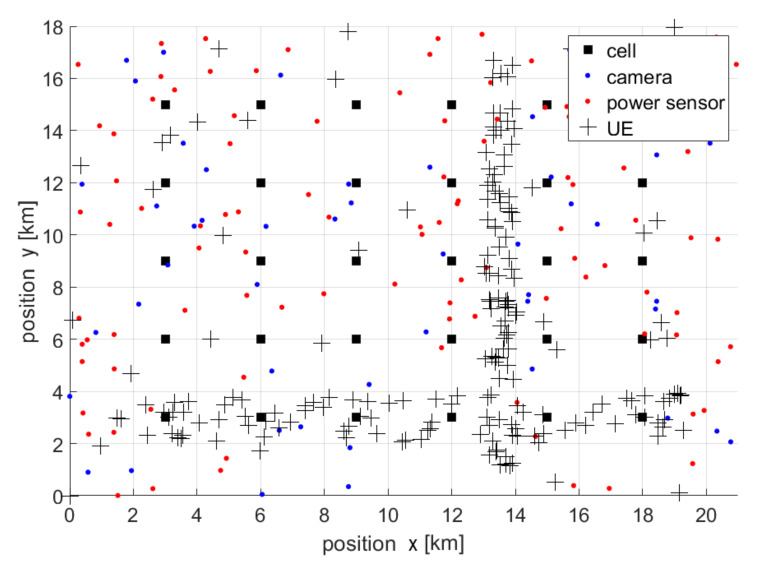
Example of a network topology.

**Figure 2 sensors-22-03515-f002:**
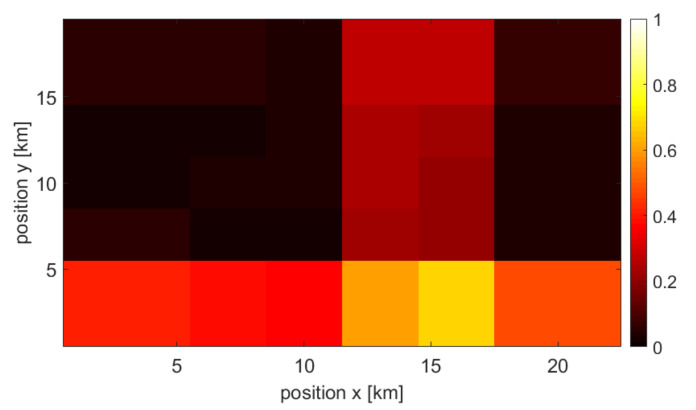
Example of a user density map created by data from access points.

**Figure 3 sensors-22-03515-f003:**
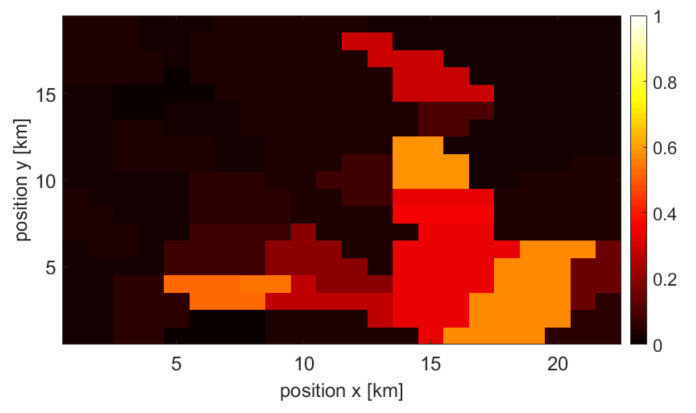
Example of a user density map created by camera data.

**Figure 4 sensors-22-03515-f004:**
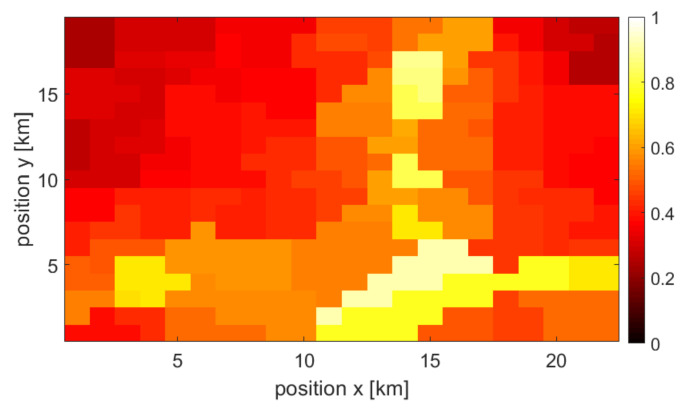
Example of a user density map created by power sensor data.

**Figure 5 sensors-22-03515-f005:**
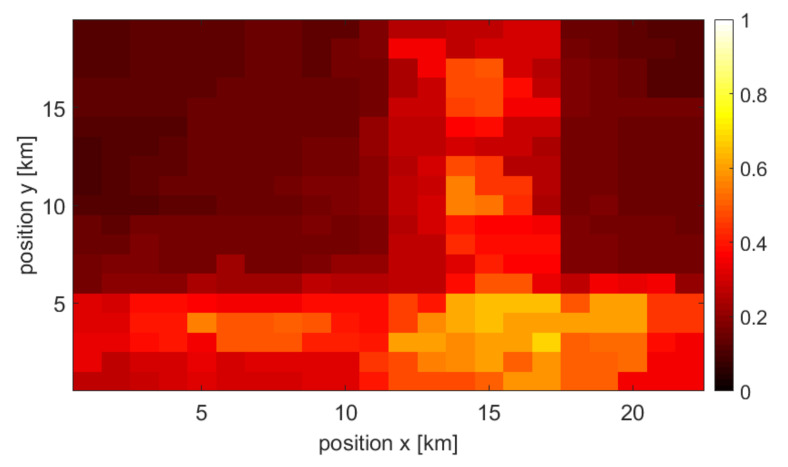
Example of a user density map merged using all three types of maps.

**Figure 6 sensors-22-03515-f006:**
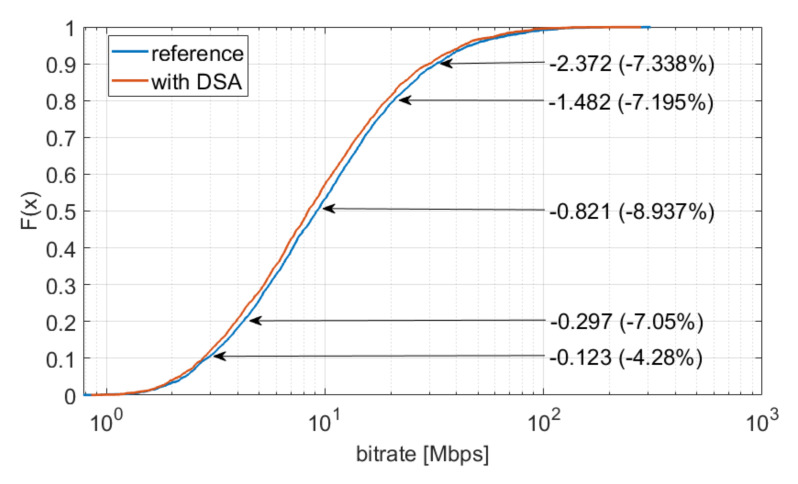
Single simulation repetition results—user bitrate.

**Figure 7 sensors-22-03515-f007:**
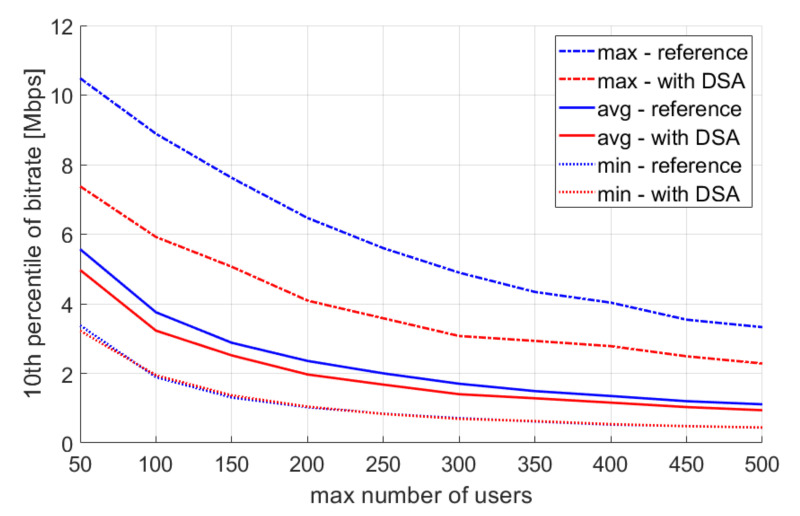
Single simulation repetition results—user bitrate.

**Figure 8 sensors-22-03515-f008:**
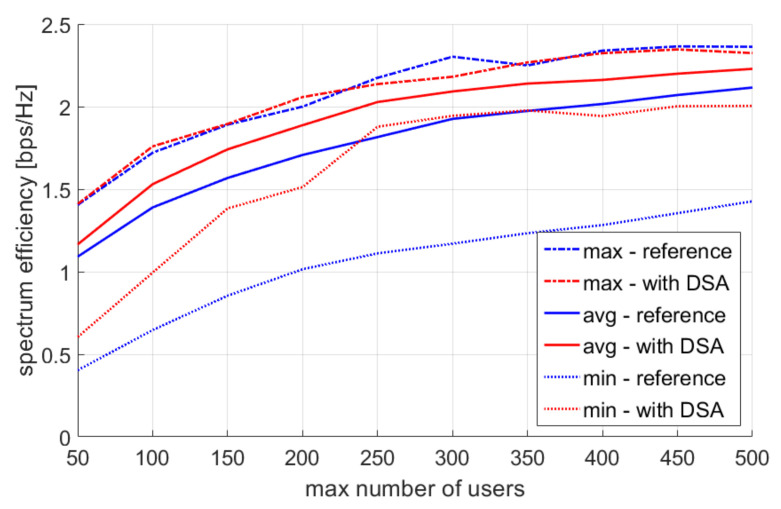
Single simulation repetition results—spectral efficiency.

**Figure 9 sensors-22-03515-f009:**
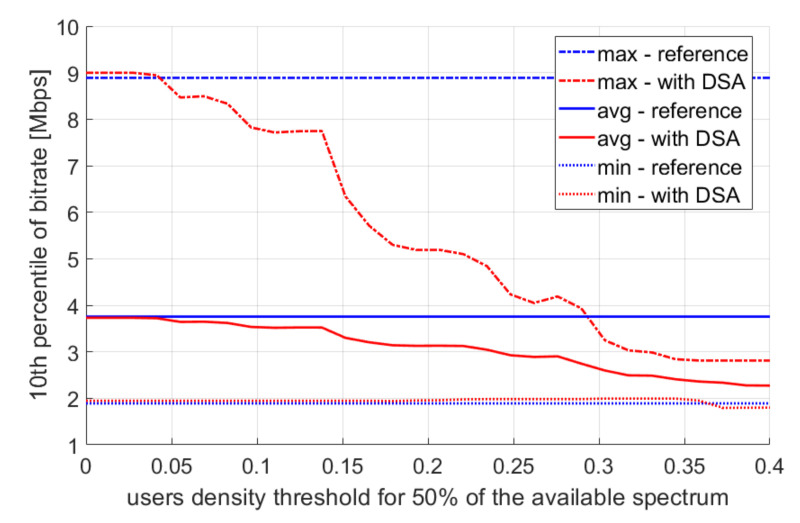
Multiple simulation repetition results—10th percentile of user bitrate.

**Figure 10 sensors-22-03515-f010:**
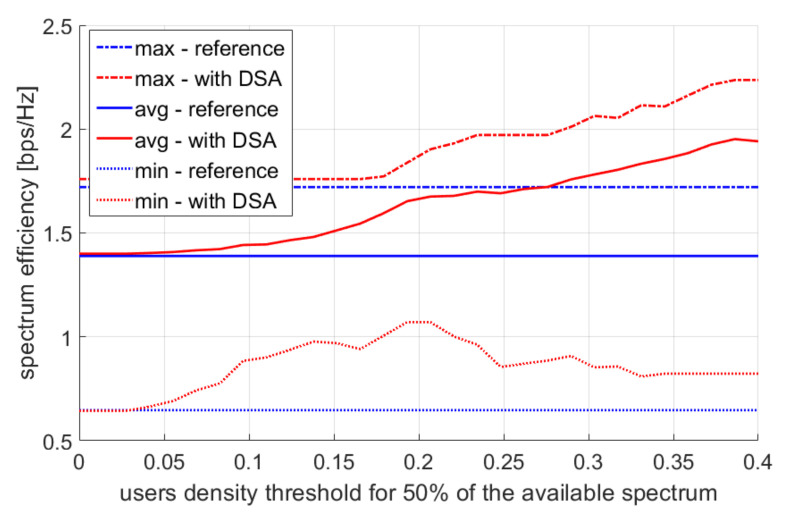
Multiple simulation repetition results—spectrum efficiency.

**Figure 11 sensors-22-03515-f011:**
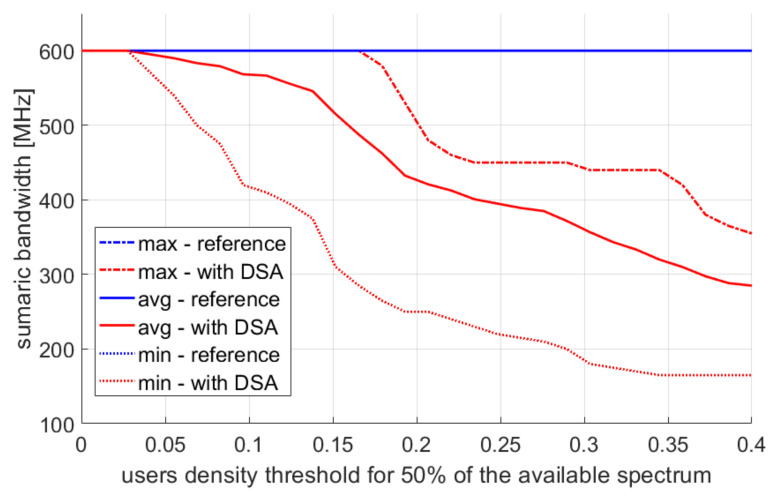
Total frequency band used in the network.

**Table 1 sensors-22-03515-t001:** Single simulation run parameters.

Parameter Name	Unit	Value
Area size	km	21.0 × 18.0
Base station spacing	km	3.0
Map grid spacing	km	1.0
Number of base stations	-	30
Max number of UEs	-	100
Max number of persons	-	300
Number of cameras	-	50
Number of WLAN APs	-	100
Number of days—$N_{days}$	-	50
Number of time stamps	-	6
Frequency	MHz	2400
Transmit power	dBm	20.0
Max bandwidth	MHz	20.0
Interpolation method	-	nearest-neighbor

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
