# Peer review of "Dynamic Spectrum Allocation Using Multi-Source Context Information in OpenRAN Networks"

_sensors, 2022, doi:10.3390/s22093515_

Round 1
Reviewer 1 Report
- The abstract should include the contributions of the results obtained
- Reference [15] does not exist in the document
- Why in the combinatorial equation (1), (x,y) is raised to the ' ?
- There are several grammatical errors in the document. For example "The example of achieved normalized maps originated from the three above-mentioned sources at specific time are" is better written like this: "The example of achieved normalized maps originating from the three above-mentioned sources at a specific time is shown in Fig. 2, Fig. 3, and Fig. 4, respectively".
- In the document is indicated: "A single iteration of the simulation consists of calculating the distance between all devices, determining the received power, assigning users to the access point with the highest received power value, allocating spectrum resources, calculating the signal-to-interference-plus-noise (SINR) value, calculating the achieved bitrate."; but it is not explained how the above was done, through references or the equations.
- Checking this sentence seems to have errors: "... than T1 than T25..."
- Policies must be justified: " the authors selected arbitrarily the following policy"
- Results can have further quantitative analysis
- Conclusions should be included in the document
Reviewer 2 Report
General comments:
The presented paper was devoted to the well-known issues related to the management and optimization of radio resources in modern radio access systems. The authors of the paper presented the results of their pilot researches in a very accessible way, which deserves a special mention. The state-of-the-art was conducted in a very synthetic manner, which causes a certain insufficiency and the suggestion that not much has been done in this field - it is worth expanding this part of the paper a bit.
Detailed comments:
The paper was written in a concise and legible manner, but requires a few minor corrections and extensions:
- Please correct the entries: “[12? –14]” – line 53, „of Ndays days” – line 221.
- The text “ is greater than T1 than T25,“ – line 213, is not clear. Please correct it according to the conditions from lines 249-250.
- Figure 1 – please insert at the markings of the “x” and “y” axes – for clarity.
- The so-called “traditional scenario” should be described in more detail.
- Please rebuild the legend in the drawings so that the max is at the top and the min at the bottom. The current form makes the analysis difficult.
- Figure 11 – why max bandwidth is lower than min bandwidth?
Summary:
The simulation results presented in the paper show that the proposed method of using additional information on potential radio devices appearing in the area of network operation brings measurable benefits in the radio spectrum management system. The presented description of the simulation results, after small adjustments and extensions, will be a good contribution to the knowledge of radio resource management techniques in open radio access networks.
